# Modified Beetle Annealing Search (BAS) Optimization Strategy for Maxing Wind Farm Power through an Adaptive Wake Digraph Clustering Approach

**Yanfang Chen [1,2], Young-Hoon Joo [2,*] and Dongran Song [3]**

1 School of Electronics and Information Engineering, Jiujiang University, Jiujiang 332005, China; 6110011@jju.edu.cn
2 School of IT Information and Control Engineering, Kunsan National University, Kunsan 54150, Korea
3 School of Automation, Central South University, Changsha 410083, China; songdongran@csu.edu.cn
* Correspondence: yhjoo@kunsan.ac.kr

**Abstract:** Owing to scale-up and complex wake effects, the centralized control that processes the command from turbines may be unsuitable, as it incurs high communication overhead and computational complexity for a large offshore wind farm (OWF). This paper proposes a novel decentralized non-convex optimization strategy for maxing power conversion of a large OWF based on a modified beetle antennae search (BAS) algorithm. First, an adaptive threshold algorithm which to establish a pruned wake direction graph while preserving the most critical wake propagation relationship among wind turbines are presented. The adaptive graph constraints were used to create wake sub-digraphs that split the wind farm into nearly uncoupled clustering communication subsets. On this basis, a Monte Carlo-based beetle annealing search (MC-BAS) nonlinear optimization strategy was secondly designed to adjust the yaw angles and axial factors for the maximum power conversion of each turbine subgroup. Finally, the simulation results demonstrated that a similar gain could be achieved as a centralized control method at power conversion and reduces the computational cost, allowing it to solve the nonlinear problem and real-time operations of the OWF.

**Keywords:** beetle antennae search optimization; wake propagation; direct graph; offshore wind farm; clustering subset; graph adaptive pruning





## 1. Introduction

With increasing environmental problems, such as the greenhouse effect, clean energy has become a critical issue that needs to be solved worldwide. Wind energy has become very competitive in comparison with other green energy production technologies for the mature control technology, such as the control approach to improve the performance of wind turbines in different conditions [1], model predictive control [2], etc. Moreover, the exploitation of wind energy is mainly fulfilled by wind turbines in arrays or groups to reduce the cost of energy [3,4]. Additionally, energy storage is also a very important branch [5].

To increase the total output power, the optimization methodology [6] plays a vital role by considering the topography, prevailing wind direction environment [7], and the space of turbines. For a designed wind farm, there are also some other methodologies for improving the total output power [8–10], decreasing the thrust load [11], improving the lifetime of turbines [12], and tracking power reference signals to improve wind integration into the power grid [13–16]. Researchers have recently proposed some control methods for optimizing power conversion in large wind farms without considering wake infection [17,18]. Some kinds of literature [19–21] propose novel graph theories. The production of generating an interaction matrix is treated in the same way. However, the difference is the kinetic energy deficits with considering the wind speed and direction probabilities, which is explained in detail in Equation (7) in paper [18]. Additionally, in this paper, we

focus on the variations of wind velocity between the upstream and downstream turbines. However, the constraints of the offshore land limit the space of turbines, not infinitely long. Therefore, the wake effects in a wind farm are inevitable, and the closely spaced turbines will produce little output power. The reason is that wake affects the downstream wind speed. Moreover, the relationship between the upstream and downstream turbines is inseparable from the basic turbine model, and the existing turbine models currently contain Jensen's model [22], Frandsen's model [23], CFD-wake model [24], etc. The Jensen and Frandsen models have assumed "top-hat" shape distributions for the wake velocity deficit. In fact, the actual distribution is Gaussian as in [21,25], which is based on self-similarity theory and is often used in free shear flows. It has become a hot topic that the optimization algorithm performs the search of the values of optimal axial and yaw angle to maximize the output power and mitigate the wake interference.

For large-scale wind farms, the burden of communication with the central unit is significantly large, and the speed of centralized implementation is no longer suited for real-time control [15,16]. More recent studies have investigated decentralized control, distributed control [26], and cooperate control [27] which are proposed to decrease the communication burden and improve control speed by dividing whole turbines into several decoupled subsets. Centralized control only suits a small-scale wind farm because there is much large information sharing between turbines and the central controller, and every turbine needs no other turbines' information. Distributed control method was proposed in [26]; there are several controllers in local groups instead of the central controller, and some key information of shared turbines will be transmitted between them. Moreover, every turbine considers the power information of itself and the neighbors to set the control actions. Additionally, a cooperative optimization problem is designed in [27], the information of the whole wind farm output power is transmitted to every turbine, and the control variable is axial induction factors.

For example, an equivalent model for wind farm reduction was proposed in [28] to cluster the same-feature wind turbines into a group. However, for a complex wake, the same feature presented a difficult problem to find the k-means clustering algorithm [29,30], k-median clustering algorithm [31], and k-nearest physical neighbor [32], which were proposed to uncover the clustered index. Moreover, the disadvantage of [33,34] was the clustering index with one-dimensional data, which only focuses on wind speed without considering the wake effect. However, for a large-scale offshore wind farms, one-dimensional data cannot explain the wake relationship matrix of the wake effect, and two-dimensional data, considering the wind speed and the wake effect are more suitable. Therefore, based on a clustering index with two-dimensional data, another possible solution was to propose a novel clustering approach that can construct decouple communication architecture. In [35], the turbines were clustered into some groups through the singular value decomposition (SVD) clustering algorithm. However, those clustering algorithms were assumed to behave similarly and were only valid for limited wind directions without considering the differences in the incoming wind.

As mentioned previously, there is a lack of efficient cluster methods to cluster the subset of communication neighbors of the turbines considering complex wake effects. To solve this problem, in this paper, we propose an adaptive pruning wake digraph to divide the large-scale offshore wind farm (OWF) into decoupled groups, then cluster the neighbors into the same subset by setting the k-median of two turbines' wake imperfect weights as the clustering index. The wind farm control and the optimization problem in every subset are non-convex, so it is critical to devise an efficient optimization algorithm. By combining the wake model and the single objective function, there are some optimization methods proposed in the literature [26,36–39]: in [38], a gradient-based method on wind farm layout optimization is presented; however, a gradient-based optimization algorithm is based on an analytical wind farm function which is simplified, so it cannot precisely reflect the wind farm control. Therefore, without the wind farm function, only some necessary measurement data for data-driven optimization algorithms are proposed, for example,

in [26,37]. However, the high model complexity of the measurable data will become significantly large. Moreover, the performance may not be good with noisy measurement data unavoidable in real applications. The authors of [36] explored a cooperative greedy algorithm optimizing energy production of wind farms; because of the wake effects, the downstream turbine could not generate much energy, so the total output power was not maximum.

Due to the large amount of large-scale wind farm data and nonlinear and non-convex characteristics, the above-mentioned control methods become unsuitable for obtaining the global optimum. Evolutionary algorithms (EAs) are one class of novel nature-inspired global optimization algorithms that are proposed in the literature [40,41]. For maximizing the total output power of wind farms, other algorithms such as modified grey wolf (GW) [42,43], particle swarm optimization (PSO) [44,45], and genetic algorithm (GA) [46] have been proposed. The authors of [44] present PSO intelligent algorithms for power conversion maximization using a nonlinear wake model. Importantly, no one algorithm can fit all systems. The above-mentioned algorithms are prone to premature convergence leading to a local optimum, not a global optimum, because of some unsuitable parameters. Therefore, we use a novel intelligent algorithm dubbed the beetle antennae search (BAS) [47] to improve calculation performance and searching ability to maximize wind farm power conversion. The BAS may generally converge early and fall into the optimal local solution for the unsuitable step size. In addition, the Monte Carlo (MC) method can be used to prevent the evolutionary algorithm from stagnating at a local optimum.

To summarize, in this paper, we first define an adaptive wake digraph to cluster a subset of communication neighbors of turbines. Secondly, we propose an MC-BAS optimization algorithm based on adaptive communication network topology to solve the non-convex power optimization problem. Finally, the yaw angles and axial factors are optimized to increase the power output by simulating a wind farm with $2 \times 2$ turbines, $3 \times 3$ turbines, and $5 \times 5$ turbines. The main contributions are summarized as follows:

- A decentralized coordination control scheme is achieved by controlling the yaw angles and axial factors to maximize power conversion on the wind farm. Large-scale wind farms are divided into several decoupled subsets, and then the local controller only controls the local subset's data. The proposed control scheme enables efficiency in the real-time application by optimizing the decentralized coordination to reduce computational burden and information exchange.
- A wake-based graph adaptive pruning approach is presented to split a large wind farm into several clustering subsets. This approach aims to find a decoupled sub-graph that can preserve essential distribution characteristics of the original wake direction graph. We adopt a graph clustering algorithm to divide turbines via wake graphs adaptive pruning constraint, and threshold $\varepsilon_k$ which is a vital point parameter to control the number of groups of the pruned wake digraph.
- We develop a modified BSA optimization algorithm based on adaptive pruned communication architectures. The Monte Carlo (MC) law of Simulate Anneal (SA) is introduced to improve the BAS, which significantly improves the reproducibility and stability of the algorithm. Finally, the improved algorithm is applied for wake steering control and maximum power conversion on the wind farm.

This study is organized as follows: Section 2 introduces the Gaussian-based wake model considering yaw angle. Section 3 introduces the algorithm to cluster turbine via adaptive pruning wake digraph through setting the suitable global threshold $\varepsilon_p$. In Section 4, the new MC-BAS control strategy in OWF is proposed to optimize the axial vale and yaw angle in every subset. Section 5 presents the simulated result of the proposed algorithm, the efficiency of power optimization and minimizing the calculating time is verified, and some important look-up tables are constructed. Finally, some important conclusions and summaries are presented in Section 6.

## 2. Gaussian-Based Wake Model Considering Yaw Angle

This section describes the wind turbine wake model for total output power optimization through axial induction and yaw angle control. The three-dimensional wind velocity deficit behind the upstream turbine $i$ is defined as Gaussian shape, which was derived through Navier–Stokes equation [25]:

$$\frac{V(x,y,z)}{V_\infty} = 1 - Ce^{-(y-\delta)^2/2\sigma_y^2}e^{-(z-z_h)^2/2\sigma_z^2} \tag{1}$$

where $V$ denotes the velocity in the wake, $V_\infty$ denotes the free-stream inflow wind velocity of the wind farm, $x, y, z$ is the direction of streamwise, horizontal spanwise, and vertical spanwise, a decoupled topology, $\delta$ is the wake centerline, $z_h$ is the hub height, $\sigma_y$, $\sigma_z$ is the wake expansion in $y, z$ direction, and $C$ is the velocity deficit at the wake center. The main parameters are shown in Appendix A.

The relationship of $\alpha$, and $\gamma$, can be defined as [21,25]:

$$\alpha \approx \frac{0.3\gamma}{\cos\gamma}\left(1 - \sqrt{1 - C_T\cos\gamma}\right) \tag{2}$$

where $C_T$ denotes the thrust coefficient.

The relationship between the initial lateral deflection of wake deflection $\delta_0$ as denied as in [25], $x_0$ is the length of the near wake as defined in [25], and the wake deflection angle $\alpha$ can be defined as the following equation [21]:

$$\delta_0 = \delta_0\tan\alpha \tag{3}$$

$$\delta = \delta_0 + \frac{\theta E_0}{5.2}\sqrt{\frac{\sigma_{y0}\sigma_{z0}}{k_y k_z M_0}}\ln\left[\frac{(1.6 + \sqrt{M_0})\left(1.6\sqrt{\frac{\sigma_y\sigma_z}{\sigma_{y0}\sigma_{z0}}} - \sqrt{M_0}\right)}{(1.6 - \sqrt{M_0})\left(1.6\sqrt{\frac{\sigma_y\sigma_z}{\sigma_{y0}\sigma_{z0}}} + \sqrt{M_0}\right)}\right] \tag{4}$$

After describing the atmospheric wake model of the wind field, the turbine model used in this paper is introduced. The turbine model consists of a power coefficient $C_p$ and thrust coefficient $C_T$ which are all based on wind speed, tip speed ratio, and a blade pitch angle. In this paper, the $C_p$ and $C_T$ curves are used to form the fitting data of FAST and the National Renewable Energy Laboratory's (NREL's) 5 MW turbine [48].

To calculate the output power $P$ of a wind turbine, the formula can be shown as follows [30]:

$$P_j(\alpha_j, \gamma_j; V_j) = \frac{1}{2}\eta\rho A_j C_P\cos(\alpha_j, \gamma_j)^{1.88}V_j^3 \tag{5}$$

where $\eta$ denotes generator efficiency; $\rho$ is the air density; $A_j$ is the rotor swept area; $\cos(\alpha_j, \gamma_j)^{1.88}$ represents the correction factor of axial factors $\alpha_j$ and yaw misalignment angle $\gamma_j$; wind velocity $V_j$ can be calculated from Equation (1).

As depicted in Equations (1)–(5), the power conversion can be optimized by adjusting the axial factors $\alpha_j$ and yaw angle $\gamma_j$. The interested reader can read a more detailed description of wake deflection in [19].

## 3. Clustering Turbine via Pruning Wake Digraph

This section will partition the large-scale OWF into several decoupled subsets based on the weights calculated from the k-Median clustering algorithm [31].

The adaptive pruning wake digraph process can be summarized in Figure 1, including the wake farm model, original digraph generation, digraph pruning, and turbines clustering. The decision variable is the magnitude of the strength between turbines $w_{ij}$ as shown in Figure 1c,e,g, then the large-scale OWF can be defined into several decoupled subsets. The section is focused on how to obtain adaptive wake digraphs of the large-scale OWF with wake interaction.

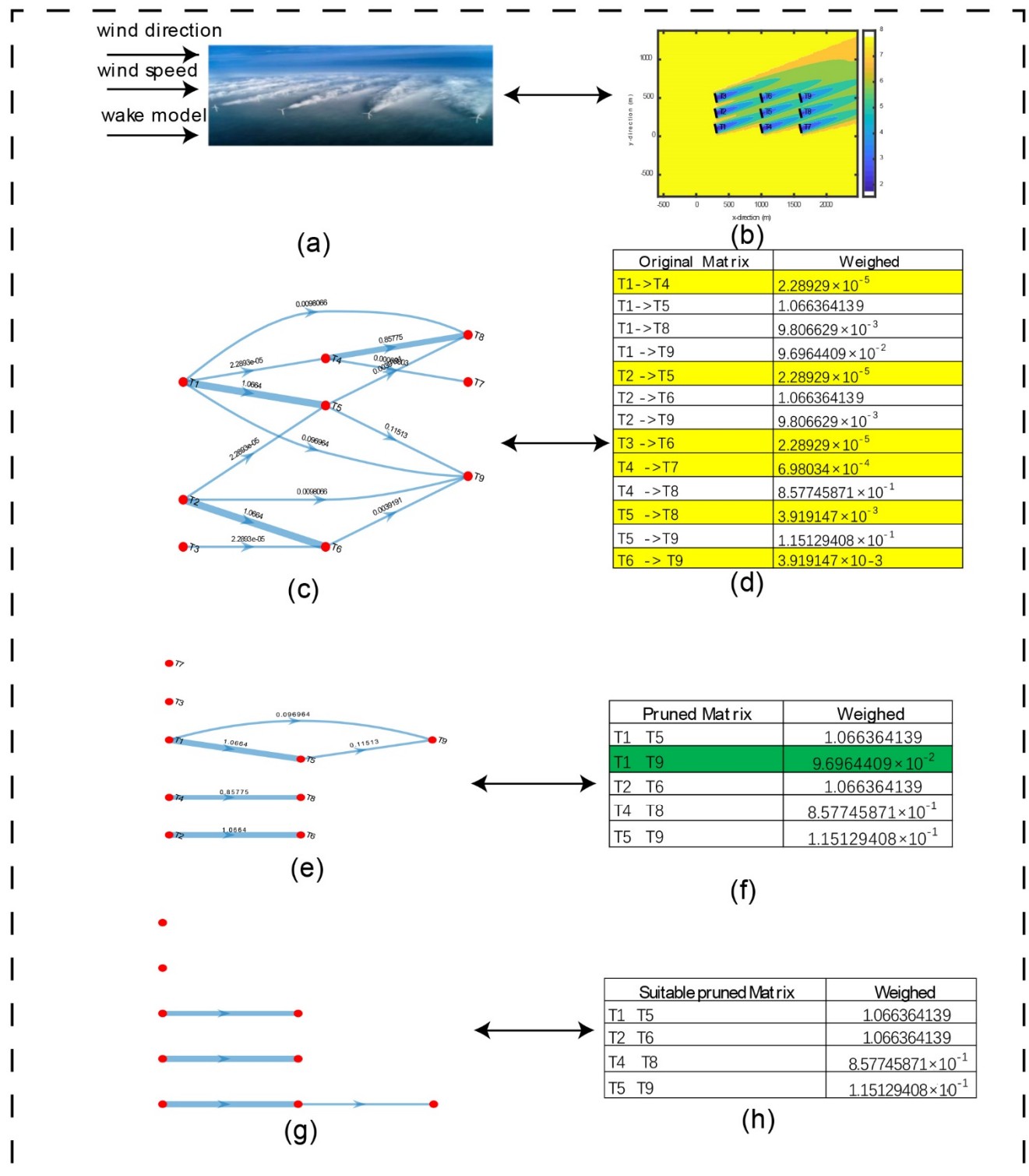

**Figure 1.** The proposed whole decentralized control scheme on a $3 \times 3$ matrix OWF with $V_\infty = 8$ m/s, $\varphi = 15°$: (**a**) Wind Farm; (**b**) wake field; (**c**) original wake digraph; (**d**) turbine original matrix; (**e**) pruned wake digraph with $k = 3.2$; (**f**) turbine pruned matrix with $k = 3.2$; (**g**) pruned wake digraph with $k = 5.1$; (**h**) turbine pruned matrix with $k = 5.1$.

### 3.1. The Original Wake Graph of Wind Farm

An autonomous wind farm can be modeled as a weighted directed graph with wake impact from upstream turbines i to downstream turbines j [49]. Under this paradigm, the digraph is determined by the wind speed direction and wind farm layout. Assuming that there are N turbines in the OWF, we will explain how to construct the original wake digraph $\mathcal{G}$ in detail.

To demonstrate the advanced approach, some definitions as follows are necessary:

**Definition 1.** *The original wake digraph $\mathcal{G} = (\mathcal{V}, \mathcal{E})$ where vertices $\mathcal{V}$, $\mathcal{V} = \{v_i i = 1, 2, \ldots, N\}$ denote the turbines. Edges $\mathcal{E}$, $\mathcal{E} \subset \mathcal{V} \times \mathcal{V}$ represents the wake distribution between every pair of upstream turbine and downstream turbine.*

$\mathcal{E}$ can also be used the weight to indicate the strength of interactions between two nodes [30]:

$$\mathcal{E} = \{w_{ij} : i, j \in \mathcal{V}\} \tag{6}$$

where $w_{ij}$ is a non-negative value. When the upstream turbine $v_i$ exerts a wake effect on the downstream turbine $v_j$, the following equation can describe it:

$$w_{ij} = \begin{cases} \dfrac{A_{overlap\ i,j} * V_{wake}}{x/D}, & \text{shadowing,} \\ 0, & \text{no shadowing.} \end{cases} \tag{7}$$

where $V_{wake} = \dfrac{V_\infty - V_j}{V_\infty}$, the wake overlap effect area $A_{overlap\ i,j}$ is described in Figure 2, $x$ represents the physical distance between the upstream turbine $V_i$ and downstream turbine $V_j$; $D$ represents the turbine rotor diameter of all the turbines. It is critical to note that the wake distribution should remain constant during the control period so that the control speed is high enough to counteract the changing wake distribution.

- The communication neighbors of vertex (turbine) $v_i$ are denoted by $N_i = \{v_j | ((v_i, v_j) \in \mathcal{E})\}$.
- The set of shared turbine $S_i$ in communication neighbors between the subset $N_i$ and subset $N_j$, are denoted as $S_i = \{T_i | T_i \in N_i \cap N_j\}$ where $T_i$ represents the shared turbine numbers in different subsets.

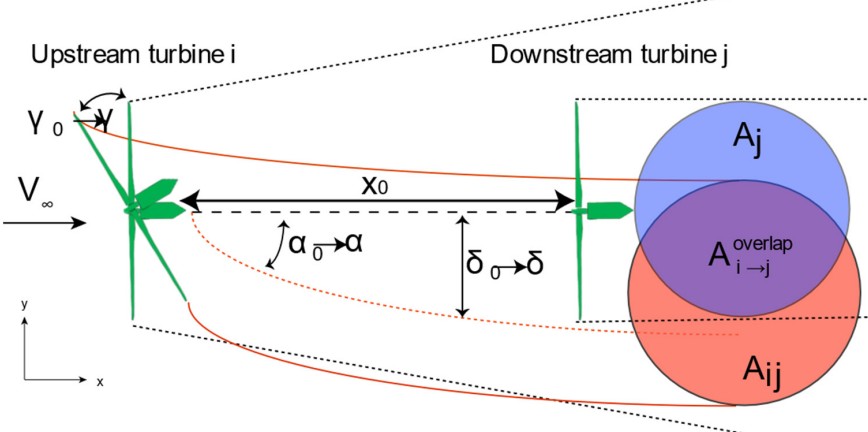

**Figure 2.** An example of a two-turbine wake redirection control through setting the yaw misalignment angle. $\gamma$ is the upstream turbine *i*'s yaw angle, $\alpha$ is the deflection angle, and $\delta$ denotes the wake deflection. The black dashed lines represent the wake of the upstream turbine *i* with no yaw control, and the red lines indicate the wake of the upstream turbine *i* with yaw control. $A_{i \to j}^{overlap}$ represents the area overlap ratio that the wake effect area $A(Ti|Tj)$ of the upstream turbine *i* to the downstream turbine j and the rotor area $A(Tj)$ of downstream turbine *j*.

*3.2. Decoupled Communication Scheme of Wind Farm Based on Adaptive Pruned Algorithm*

For most turbines, the coupling degree of one turbine with other turbines may be high or low. Therefore, the pruned wake digraph $\mathcal{G}_p$ can be pruned from the original wake digraph $\mathcal{G}$.

**Definition 2.** *The pruned wake digraph* $\mathcal{G}_p = (\mathcal{V}, \mathcal{E}_p,)$, *pruning edges* $\mathcal{E}_p$ *set* $\mathcal{E}_p \subset \mathcal{V} \times \mathcal{V}$, $\mathcal{E}_p = \{ w_{ijp} : i, j \in \mathcal{V} \}$
*where*

$$w_{ijp} = \begin{cases} w_{ij} & w_{ij} \geq \varepsilon_k, \\ 0, & w_{ij} < \varepsilon_k \end{cases} \tag{8}$$

- The adaptive threshold $\varepsilon_k = k * \varepsilon$, $k$ is one hyper-parameter, and $\varepsilon$ is the basic threshold.
- Basic threshold $\varepsilon$ is defined as the geometric median of the whole wake weight coefficients. The central idea of the geometric median is as follows: given the set of $n^2$ points $w_{11}, w_{12} \ldots, w_{ij}, \ldots, w_{nn}$ find a value $\varepsilon$ that minimizes the sum of Euclidean distance:

$$f(x) \stackrel{\text{def}}{=} \sum_{i=1}^{N} \sum_{j=1}^{N} \| x - W_{ij} \|_2 \tag{9}$$

where $\varepsilon \epsilon \ argmin(f(x))$.

A decoupled topology in this section will be achieved by using the adaptive pruned wake digraph algorithm. However, determining the degree of pruning digraph has not been deeply studied, and there is little literature discussing it. This paper proposes an adaptive pruned algorithm to find a suitable threshold $\varepsilon_k$ to solve the grouped problem by obtaining more reasonable decoupled subsets.

Based on the pruned wake digraph $\mathcal{G}_p = (\mathcal{V}, \mathcal{E}_p)$, for each angle $\varphi \in \{\varphi_1, \ldots, \varphi_w\}$, there are clustering subsets $N_l, l \geq 1$. A given direction $\varphi$ has a corresponding cluster subset $N_i \in \{N_1, \ldots, N_M\}$, $M$ is the number of subsets. Then, according to the wind farm layout, we build the original wake digraph $\mathcal{G}$ and calculate the weight coefficient matrix $w_{ij}$ to find the communication neighborhood of turbines. The algorithm for turbine clustering via adaptive pruned wake digraph is shown as follows (Algorithm 1):

---

**Algorithm 1:** The method of clustering turbine via pruning wake digraph (Adaptive pruned wake digraph algorithm)

---

**Step 1:** Based on the layout of the position of the wind farm $(X, Y)$, collect all relevant parameters, including wind direction $\Phi$, wind speed $V_\infty$. Additionally, the parameters of the wind turbines, for example, the rotor diameter $D$, the physical distance $x$ between WTs, and the overlap wake area $A_{i \rightarrow j}^{\textbf{overlap}}$, etc.

**Step 2:** Calculate the threshold $\mathcal{E}$, and set the initial hyper-parameter $k$, step hyper-parameter $\Delta k$.

**Step 3:** Obtain the pruned digraph $\mathcal{G}_p$ from the original wake digraph $\mathcal{G}$ according to the global threshold $\varepsilon_k$ according to the global threshold $\varepsilon_k$.

**Step 4:** Digraph clustering. Firstly, define the leading turbines for each subset that is experiencing free-stream velocity $V_\infty$. Secondly, each leading turbine decides the communication neighbors through the connectivity information of the digraph $\mathcal{G}_p$ by a depth-first tree search (BFS) algorithm into the same subset $N_i$.

**Step 5:** If there is a set of shared turbines $S_i$, we need to continue to tune the value of k by set $k = k + \Delta k$, go back to Step 3. If not, go directly to step 6.

**Step 6:** Calculate the output power and calculating time with the $k$ value from step 5, save the parameters.

**Step 7:** If the coefficients of $\mathcal{G}_p$ are not all 0, continue to tune the $k$ value by setting $k = k + \Delta k$, go back to Step 2. If the coefficients of $\mathcal{G}_p$ are all 0, go to step 8.

**Step 8:** Based on the adaptive pruned wake digraph $\mathcal{G}_p$, we can establish turbine clustering subsets $N_i$ and analyze all the parameters with different $k$, and select the suitable value $k_2$.

---

Overall, the adaptive pruned digraph and the contribution can be simplified as follows: Firstly: for one, considered wind speed and wind direction, to satisfy the pruned wake digraph are decoupled (no shared wind turbines between all the subsets), the range of $k$ can be found out to be $k \in [k_1, k_3]$, $k_1$ is the minimum value, and $k_3$ is the maximum value

of *k*. In most literature, the *k* is one random value between the range so that the result may be suboptimal.

Secondly: by comparing the performance of output power and calculating time, deduce a suitable value $k_2$.

Thirdly: change the wind speed and direction by the above method and obtain another corresponding suitably $k_2$.

Lastly: create a line-off query look-up table using the obtained result, which can be used for quick reference with input winds velocity $V_\infty$ and wind direction $\varphi$, the output is $k_2$.

The contributions are mainly to select suitable parameter $k_2$ with the proposed algorithm. Based on this condition, the control optimization can obtain the best result. The procedure will be demonstrated in Section 5 by one simulation example.

## 4. Wind Farm Control Strategy

Upon establishing the pruned wake digraph and clustering subsets, the overall OWF is controlled by multiple individual independents rather than by a single controller. The decentralized optimization process is described in this section. In order to achieve the control objectives of real-time output power optimization, a decentralized control scheme is proposed for the large-scale OWF, as shown in Figure 3. This control scheme is divided into two steps. First, wind farms are decoupled into several independent wind turbine clusters, and their communication neighbors are determined by the network topology of the adaptive pruned wake digraph. Therefore, a decentralized control strategy for OWF is proposed to realize the power control of the host computer. Second, we use the beetle antennae search (BAS) algorithm approach to optimize the yaw setting and the axial induction factor setting in the OWF to maximize the total wind plant power conversion.

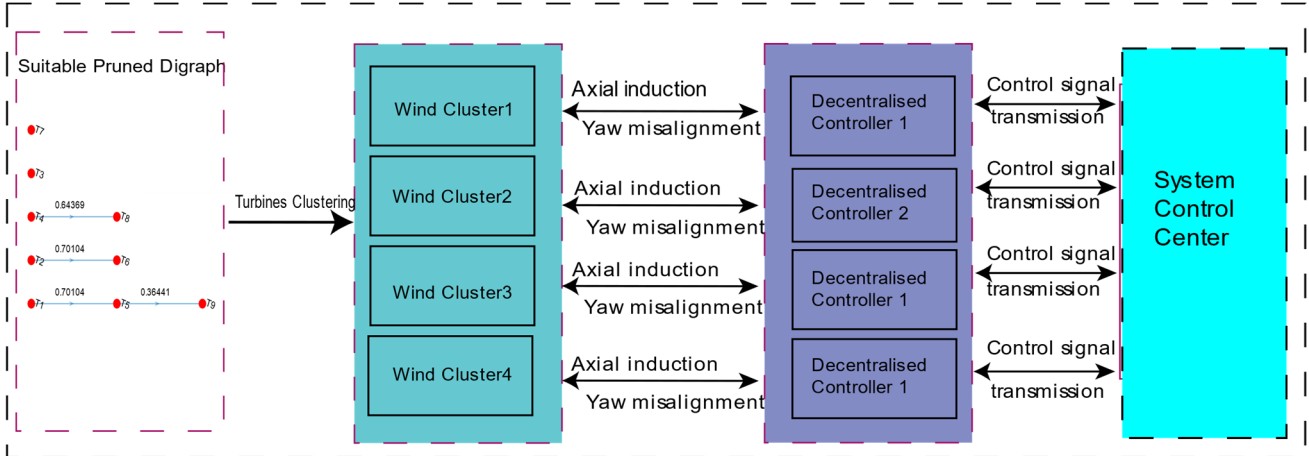

**Figure 3.** The proposed whole decentralized control scheme on a 3 × 3 turbines offshore wind farm.

### 4.1. The Output Power Optimization Problem

For a given decoupled clustering subsets $N_i$, the corresponding decentralized power control is based on the optimization objective: minimizing the power output of wind farms in Equation (4). Each wind turbine cluster is composed of several neighbor wind turbines that are decentralized on different communication network topologies of adaptive pruned wake digraphs. The proposed decentralized control scheme is shown in Figure 3. The corresponding decentralized power control for a given clustering subset $N_i$ is based on the following single-objective optimization problem. The control variables for the optimization

are the axial factor $\alpha$ and yaw angle $\gamma$ of the wind turbines. The whole OWF decentralized optimization function $f(x)$ can be expressed as follows:

$$\min_{x} f(x) = -\sum_{i=1}^{M} \sum_{j=1}^{K} \left( P_{i,j}(\gamma_{i,j}, \alpha_{i,j}; V_{i,j}) \right)$$

$$\text{s. t.} \begin{cases} -\gamma_{min} \leq \gamma_{i,j} \leq \gamma_{max} \\ \alpha_{min} \leq \alpha_{i,j} \leq \alpha_{max} \end{cases}$$

(10)

where $i$ indicates subsets, j indicates turbines in subsets, M indicates the number of uncoupled subsets on the wind farm, and K indicates the number of turbines in every subset. The yaw angle $\gamma_{i,j}$ is between the upper bound $\gamma_{max}$ and the lower bound $-\gamma_{min}$; $\alpha_{min}$ and $\alpha_{max}$ represent the lower bound and upper bound range of axial factor $\alpha_{i,j}$ and the power $P_{i,j}$ is between the minimum power $P_{low}$ and the rated power $P_{rate}$.

### 4.2. Monte Carlo Law with the BAS(MC-BAS) Controller for Wind Turbines

The OWF cost function is given in Equation (9), which is a nonlinear and non-convex optimization problem. Many methods do not guarantee to find the optimal global solution. Herein, we adopt the BAS algorithm to solve optimization problems. Like genetic algorithms (GA), particle swarm optimization (PSO), and other intelligent algorithms, BAS does not require prior knowledge of the specific shape of the function or gradient information to optimize efficiency. The main two steps are the searching process and the result detection, tuning the adaptive step size until the optimization value is reached. The advantage of BAS is simple and fast speed to get the optimization object than other intelligence algorithms. Moreover, the high-speed advantage of BAS over the particle swarm optimization algorithm is that it requires only one individual, a longicorn beetle.

The Monte Carlo (MC) law of the annealing algorithm (SA) is introduced to improve the repeatability and stability of the algorithm. The improved algorithm is then applied to wake steering control so as to maximize power conversion on the wind farm. The optimal target value of the object is determined by simulating the annealing process, and the lowest energy of the target and simulated annealing incorporates random variables during the search process. For example, it embraces a worse solution than the current solution with a certain probability, increasing the possibility of exiting local optimization. The Modified MC-BAS algorithm is shown below (Algorithm 2).

1.  Random direction vector
    To simulate the search behavior of longicorn, its direction vector is defined as [47]:

$$\vec{b} = \frac{\text{rand}(k, 1)}{\| \text{rand}(k, 1) \|}$$

(11)

    where $\text{rand}(k, 1)$ denotes a random function, and $k$ represents position dimensions.
2.  The coordinate of both right-hand and left-hand sides of the antennae of beetles are presented as [47]:

$$x_r = x^t + d^t \vec{b}$$

$$x_l = x^t - d^t \vec{b}$$

(12)

    where $t$ represents the number of iterations; $x_r$ and $x_l$ denote the spatial position of the right and left beetles of longicorn beetles in the $t$ iteration, respectively; $d^t$ represents the exploitability of antennae sensing length in the $t$ iteration.
3.  Fitness value:

$$\begin{cases} f_{right} = f(x_r) \\ f_{left} = f(x_l) \end{cases}$$

(13)



where $f_{right}$ and $f_{left}$ denote the fitness value of the right beetle and the left beard in the current spatial position; $f(\cdot)$ is the fitness function as Equation (9).

---

**Algorithm 2:** The grouped MC-BAS methodology for wind farm power production (MC-BAS Algorithm)

---

**Result:** The best yaw angles and the best axial factors $x_n^{bst}$ and the best output power $f_n^{bst}$.
**Input:** Establish output objective function $f(x_n^t)$, where variable $x_n^t = [x_1^t, x_2^t, \dots, x_N^t]$ and initialize the parameters $x^0, d^0, \delta^0, \eta_d, \eta_\delta, M_T, N, \alpha, \delta_{\text{criterion}}, t_{max}$.
**While** ($n < N$) **do**

    **While** ($t < t_{max}$) or ($\delta < \delta_{\text{criterion}}$ )**do**

        Generate the direction vector $\vec{b}$ via Equation (9);

        Search in variable space with two kinds of antennae $x_r$ and $x_l$ via Equation (12);

            Update fitness value $f(x_r), f(x_l)$ via Equation (13);

            Update the state variable $x^t$ via Equation (14);

                **if** $f(x_n^t) < f_n^{bst}$ then

                    $x_n^{bst} = x_{n'}^t$

                    $f_n^{bst} = f(x_n^t)$,

                **else**

                  delta=$(f(x_n^t) - f_n^{bst})/f_n^{bst}$;

                  $L_P$= exp (−delta / $M_T$);

                    if      **r**and (1) <= $L_p$,then

                        $x_n^{bst} = sample(x_n^t)$,

                        $f_n^{bst} = (f(x_{sample}^t))$,

                        $M_T = \alpha * M_T$

                  End

              End

        Update sensing diameter $d$,step $\delta$ via Equation (16)

    End

    Update $t$: $t = t + 1$

    Update $n$: $n = n + 1$

End

---

4.     Pre-update position:

$$x^t = x^{t-1} + \delta^t \vec{b} \, sign\left(f_{right} - f_{left}\right) \tag{14}$$

Pre-update the position of the beetles based on the iteration, and the $sign(\cdot)$ is a symbol function; $\delta^t$ is the step size factor of the algorithm in the $t$ iteration.

5.     Accepted solution using the Monte Carlo law

The Monte Carlo law of the SA algorithm is embedding into BAS. In the iterative process, the probability $P$ is used to accept the inferior solution to improve the global optimization ability of BAS:

$$L_p = \begin{cases} 1, & f(x^t) < f(x^{t-1}) \\ \exp\left(-\frac{f(x^t)-f(x^{t-1})}{M_T}\right), & f(x^t) \geq f(x^{t-1}) \end{cases} \tag{15}$$

where $f(x^t)$ denotes a pre-update position, $f(x^{t-1})$ denotes the best position in the last iteration; exp (.) represents the exponential function; $M_T$ is the higher temperature.

6.  Step size:

$$d^t = \eta_d d^{t-1} + d^0$$
$$\delta^t = \eta_\delta \delta^{t-1} + \delta^0 \tag{16}$$

where $d$ and $\delta$ denote antennae length and step size, $d^0$ and $\delta^0$ is the initial value, where $d^t$ and $\delta^t$ is the step size factor of the algorithm in the $t$ iteration, and the two parameters $\eta_d$ and $\eta_\delta$ are set by the user.

The Monte Carlo law is mixed with the BAS algorithm. The basic steps of the grouped MC-BAS algorithm can be summarized as the pseudo-code shown in MC-BAS algorithm.

## 5. Validation and Discussion

Due to the randomness and intermittence, there are no constant wind speed and constant wind direction. For simplification, based on the probability of the known wind rose, we can calculate the average value of them in 10 min, then obtain an approximately constant value to describe them. In this experiment, assume that the average wind speed $V_\infty = 8$ m/s, the range of wind direction is $\varphi = \{0°, 15°, 30°, \ldots, 180°\}$, with which the baseline direction is the $x$-axis direction and under the assumption that it is constant within one control cycle.

The reason is that the wake is affected not much by the wind speed $V_\infty$ but by the wind direction $\varphi$, so we only study the one wind speed $V_\infty = 8$ m/s, however, the whole wind direction is $\varphi \in [0°, 360°]$, with considering the symmetricity in the square wind farm, we only need to study the wind direction $\varphi \in [0°, 180°]$, and in this Simulink, we choose 5° as the step size in wind direction.

The wake digraph is the basic digraph from the wake field, as shown in Figure 4. The performance of proposed optimization approaches will be verified in this section by simulation results with the same NREL-5 MW Type III WT [48] turbine. The main parameters are given in Table A1 (Appendix A). The layout structure of OWF with lateral distance X = 500 m, longitudinal distance Y = 200 m, and rotor diameter D = 126 m, and nominal power P = 5 MW. The test is conducted for a 10-minute average of free wind speed and the direction range belongs to $\varphi \in [0°, 180°]$ at 15° increment. Furthermore, to verify the scalability and the feasibility of the proposed algorithm, in this paper, we study three different scales of wind farms with the different numbers of turbines N = 4, N = 9, and N = 25, respectively. The initial yaw angles $\gamma$ are set to 0 with a range of $\gamma \in [-30°, 30°]$, and the initial factors are set to 1/3 with a range of $\alpha \in [0, 1/3]$. It was observed that the numerical results showed that the proposed control method could reach an improved increase rate with a larger wind farm by comparing the result of $2 \times 2, 3 \times 3, 5 \times 5$ matrix turbine wind farms. In other words, verification of the proposed method means that it is suited for a large-scale wind farm.

### 5.1. Processing the Adaptive Pruning Wake Redirect Digraph

In this section, the cluster method splits the large OWF into several independent subsets using the proposed pruned wake digraph clustering approach (see Section 3). The $5 \times 5$ wind farm location under different wind directions is shown in the proposed wake digraph. In this case, we consider a wind direction of $\Phi = 45°, 90°$, the wake original digraph $\mathcal{G}$, pruned digraph $\mathcal{G}_p$ as illustrated in Figures 5 and 6, respectively. In addition, the decoupled communication topology comes from the suitably pruned wake digraph.

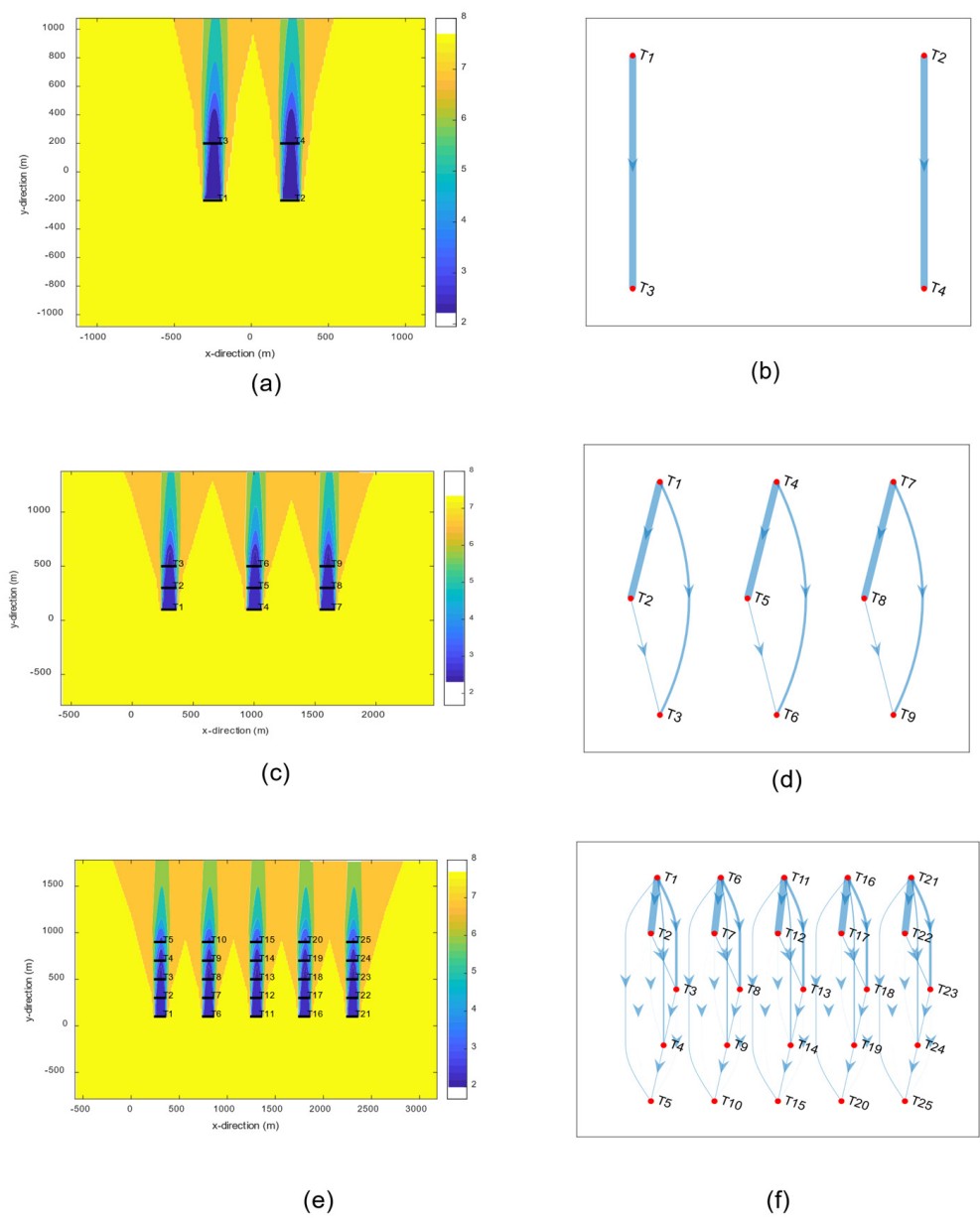

**Figure 4.** The original wake digraph $\mathcal{G}$ under different wind turbines $\varphi = 90°$: (**a**) $2 \times 2$ turbines wake field; (**b**) $2 \times 2$ turbines original wake digraph; (**c**) $3 \times 3$ turbines wake field; (**d**) $3 \times 3$ turbines original wake digraph; (**e**) $5 \times 5$ turbines wake field; (**f**) $5 \times 5$ turbines original wake digraph.

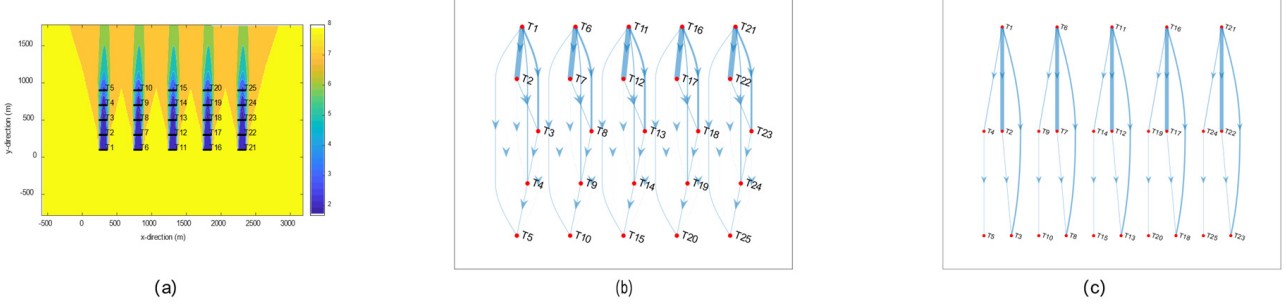

**Figure 5.** The wake digraph ($\varepsilon = 0.157481474$) with wind direction $\varphi = 90°$: (**a**) wake field; (**b**) original wake digraph ($k = 0.1$); (**c**) pruned wake digraph ($k = 3.2$).

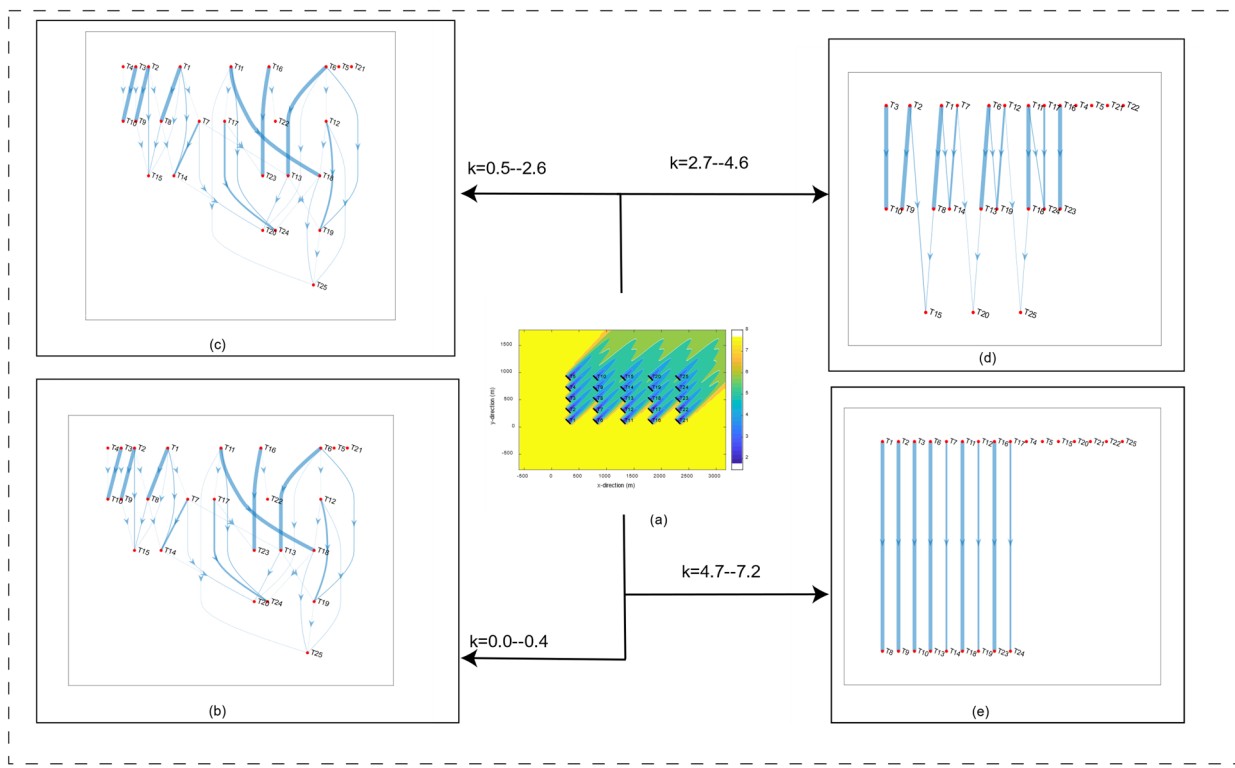

**Figure 6.** The process of adaptive pruning wake digraph method with varying $k$ for the wake digraph ($\varepsilon = 0.0122$) with wind direction $\varphi = 45°$: (**a**) wake field; (**b**–**e**) pruned wake digraph $\mathcal{G}_p$ with different $k$.

Figure 5 shows that the original wake digraph is the same as the pruned wake digraph. This is because when the wind direction is $\varphi = 90°$, the entire wake effect is concentrated on the downstream turbines without any diffused. However, when the wind direction changes to $\varphi = 45°$, the power conversion of the OWF is increased since the influence of wake interaction becomes low.

As shown in Figure 6, the original wake digraph $\mathcal{G}$ differs in the adaptive pruned wake digraph $\mathcal{G}_p$. The pruned wake digraph $\mathcal{G}_p$ is different with varying $k$. From Figure 6b to Figure 6c, the threshold $\varepsilon_p = k*\varepsilon$, $k \in [0.5, 2.6]$, so the edges which are smaller than $\varepsilon_p$ are cut off, such as the edges $\mathcal{E}_{11,25}$, etc. From Figure 6c to Figure 6d, the global threshold $\varepsilon_p$ become bigger $k$, with the range of $k \in [2.7, 4.6]$, then the edges which are smaller than $\varepsilon_p$ are cut off, for example, the edge $\mathcal{E}_{1,15}$, $\mathcal{E}_{16,25}$, $\mathcal{E}_{22,25}$, etc. Lastly, from Figure 6d to Figure 6e, get bigger $k$ at the range of $k \in [4.7, 7.2]$, the more edges are cut off, for example, the edges $\mathcal{E}_{13,20}$, $\mathcal{E}_{18,25}$, etc. and then the wake field is divided into 13 subsets with no shared turbine.

Figure 6 illustrates that the wake topology is parameter-dependent, as the external relevant variables $(\varphi, k)$ influence the wake effect. For a given $\varphi$, the suitable $k$ is vital for pruning the wake digraph to obtain the optimization decoupled subsets. The method regarding tuning the hyperparameters $k$ will be discussed in the next step, which is shown in Tables 1 and 2 as follows.

**Table 1.** The relationship of subsets and shared turbines (ST) with different $k$.

| $k$ | No of Subsets | With ST or Not |
|---|---|---|
| 0–0.4 | 9 | Yes |
| 0.5–2.6 | 13 | Yes |
| 2.7–4.6 | 13 | No |
| 4.7–7.2 | 16 | No |
| 7.3 | 21 | NO |

**Table 2.** Comparing the results of different pruning wakes under different $k$.

| $k$ | Baseline Power | Total Power (W) | Groups | T(s) | $\Delta$P |
|---|---|---|---|---|---|
| 2.7 | 3.23E+07 | $3.62 \times 10^7$ | 13 | $2.84 \times 10^2$ | 12.19% |
| 2.8 | 3.23E+07 | $3.62 \times 10^7$ | 13 | $2.44 \times 10^2$ | 12.19% |
| 3.3 | 3.23E+07 | $3.62 \times 10^7$ | 13 | $2.42 \times 10^2$ | 12.19% |
| 3.8 | 3.23E+07 | $3.62 \times 10^7$ | 13 | $2.36 \times 10^2$ | 12.19% |
| 4.3 | 3.23E+07 | $3.58 \times 10^7$ | 13 | $2.35 \times 10^2$ | 10.95% |
| 4.8 | 3.23E+07 | $3.58 \times 10^7$ | 13 | $2.32 \times 10^2$ | 10.95% |
| 5.3 | 3.23E+07 | $3.51 \times 10^7$ | 13 | $2.30 \times 10^2$ | 8.78% |
| 5.7 | 3.23E+07 | $3.43 \times 10^7$ | 13 | $2.26 \times 10^2$ | 6.30% |
| 5.8 | 3.23E+07 | $3.03 \times 10^7$ | 16 | $3.57 \times 10^2$ | −6.10% |
| 6.3 | 3.23E+07 | $3.03 \times 10^7$ | 16 | $3.59 \times 10^2$ | −6.10% |
| 6.8 | 3.23E+07 | $3.01 \times 10^7$ | 16 | $3.66 \times 10^2$ | −6.72% |
| 7.3 | 3.23E+07 | $3.01 \times 10^7$ | 21 | $3.66 \times 10^2$ | −6.72% |

Table 1 shows that there are shared turbines in the subsets depending on the value of $k$. In this paper, we focus on the range of $2.7 \leq k \leq 7.3$ because of no shared wind turbine. In other words, the subsets are all decoupled. Then, there is another problem of how to set the suitable value $k$. In this paper, the proposed adaptive pruning algorithm can solve this problem. An adaptive threshold $\varepsilon_p$ can be proposed by comparing the output power and calculating time, and the comparison results are displayed in Table 2. It is essential to note that in Table 2, considering the objective of the real-time control, we choose the suitable value $k_2$ that focuses more on computational efficiency and an increased power rate by more than 4%. Moreover, when $k$ is set as 5.3 as in Table 2, the control time is 226 s which is smaller than others. The high control speed is a very vital parameter during the control process. Therefore, we can find the suit $k_2 = 5.3$.

From Table 2, we chose $k = 5.7$ as a suitable value. The reason is that when $k > 5.7$, the output power is smaller than the baseline value, which is not permitted in this paper. Moreover, the calculation time is the smallest at the range of $0 \leq k \leq 5.7$. The controller speed is important for the objection of real-time control. In this condition, the pruned wake digraph will be divided into 13 decoupled subsets, and the clustered turbines' neighbor turbines of every subset are N1 = {T1, T8, T15}, N2 = {T2, T9}, N3 = {T3, T10}, N4 = {T4}, N5 = {T5}, N6 = {T6, T13, T20}, N7 = {T7, T14}, N8 = {T11, T18, T25}, N9 = {T12, T19}, N10 = {T16, T23}, N11 = {T17, T24}, N12 = {T21}, N13 = {T22}.

Using the above-mentioned method, the range of wind direction extends to $0° \leq \varphi \leq 90°$ with an increment of $15°$. Under different wind directions, to obtain decoupled communication topology by pruning the wake digraph, the experimental range of $k$ is $k_1 \leq k \leq k_3$ and $k_2$ is the suitable value, which can be obtained from the proposed adaptive pruning algorithm.

In this way, wind speed keeps $V_\infty = 8$ m/s, we can also obtain the suitable value $k_2$ when wind direction $\varphi$ varies from the range of [ $0°, 90°$ ] which is shown in Table 3 as follows. When wind speed $V_\infty$ and wind direction $\varphi$ changed, a look-up table of $k_2$ can be obtained by the proposed adaptive algorithm, which is shown in Appendix B—Table A2.

### 5.2. The Combined Evaluation of the Decentralized MC-BAS Algorithm

When $V_\infty = 8$ m/s, $\varphi = 45°$, and $k = 5.7$, as shown in Tables 1 and 2, the OWF wake digraph can be divided into 13 decoupled subsets. Taking the subset N2, for example, it concludes two neighbor turbines in cluster N2, the upstream turbine WT2 and the downstream turbine WT9. In this paper, the control actions and wake infection only work in the same subset. To maximize the output power of OWF, the yaw angles $\gamma$ and the axial factors $\alpha$ are activated in an optimally decentralized manner. We will explain the sensitive relationship between the control actions $\alpha$, $\gamma$, the output power $P$, and the consequent wind speed direction $\varphi$ of the neighbor wind turbine in one subset as shown in Figure 7.

Taking subset N2, for example, in the range of wind direction $\varphi \in [90°, 180°]$, the upstream turbine WT2, the output power of MC-BAS is larger than that with the greedy method as shown in Figure 7b,c. Moreover, for the downstream WT9, the output power of MC-BAS is larger than that of the greedy method. However, the reason is not because of controlling the axial factors $\alpha$ and the yaw angle $\gamma$ of WT2 but the decreasing wake effect of WT2, which is clearly shown in Figure 7e,f. For WT9 having no downstream turbines, the control parameters do not need to change the value significantly. This method is also applicable to other wind direction ranges and some other subsets. For brevity, we will not repeat the description in this paper.

When the wind direction is in the range of wind direction $\varphi \in [20°, 50°]$, WT2 and WT9 are in the same subset. In other words, WT2 and WT9 are neighbor turbines. Essentially, when wind direction $\varphi$ changes significantly, the communication topology will also vary. The wind turbine will infect each other for the same subset, and for different subsets, the wind turbines are all independent. In this way, every wind turbine in all subsets with varying wind direction is optimized, allowing the total output power to reach the maximum value. The result is shown in Figure 8 and Table 4.

**Table 3.** The different adaptive $k$ with the varying wind direction $\varphi$.

| $\varphi$ | $k$ | P(W) | $\Delta$P | T(s) |
|---|---|---|---|---|
| $\varphi = 0°$ | Baseline | $2.4556 \times 10^7$ | 0% | 0.1896 |
| | $k_1 = 0.1$ | $2.8528 \times 10^7$ | 16.18% | 276.245 |
| | $k_2 = 5.6$ | $2.8526 \times 10^7$ | 16.17% | 211.5501 |
| | $k_3 = 11.7$ | $2.0173 \times 10^7$ | −17.85% | 243.3781 |
| $\varphi = 15°$ | Baseline | $2.4973 \times 10^7$ | 0% | 0.1659 |
| | $k_1 = 1.6$ | $2.8554 \times 10^7$ | 14.34% | 268.4627 |
| | $k_2 = 2.4$ | $2.8152 \times 10^7$ | 12.73% | 138.1018 |
| | $k_3 = 2.7$ | $2.1252 \times 10^7$ | −14.90% | 189.6079s |
| $\varphi = 30°$ | Baseline | $3.1375 \times 10^7$ | 0% | 0.1595 |
| | $k_1 = 1.9$ | $3.3643 \times 10^7$ | 7.23% | 276.8732 |
| | $k_2 = 3.5$ | $3.2142 \times 10^7$ | 2.45% | 239.3284 |
| | $k_3 = 4.1$ | $2.9763 \times 10^7$ | −5.14% | 293.1692 |
| $\varphi = 45°$ | Baseline | $3.9268 \times 10^7$ | 0% | 0.1402 |
| | $k_1 = 2.5$ | $4.1118 \times 10^7$ | 4.57% | 284.4385 |
| | $k_2 = 5.7$ | $4.0926 \times 10^7$ | 4.16% | 226.5321 |
| | $k_3 = 7.3$ | $3.8126 \times 10^7$ | −2.97% | 366.5429 |
| $\varphi = 60°$ | Baseline | $3.0271 \times 10^7$ | 0.00% | 0.1385 |
| | $k_1 = 0.9$ | $3.2014 \times 10^7$ | 5.76% | 259.6893 |
| | $k_2 = 6.8$ | $3.1139 \times 10^7$ | 2.87% | 271.8649 |
| | $k_3 = 7.9$ | $2.7853 \times 10^7$ | −7.99% | 350.6543 |
| $\varphi = 75°$ | Baseline | $2.3257 \times 10^7$ | 0% | 0.1243 |
| | $k_1 = 0.6$ | $2.5473 \times 10^7$ | 9.53% | 174.9643 |
| | $k_2 = 16.5$ | $2.4385 \times 10^7$ | 4.85% | 136.9856 |
| | $k_3 = 27.3$ | $2.1072 \times 10^7$ | −9.39% | 181.6532 |
| $\varphi = 90°$ | Baseline | $1.8731 \times 10^7$ | 0% | 0.1133 |
| | $k_1 = 0.0$ | $2.2795 \times 10^7$ | 21.70% | 112.5742 s |
| | $k_2 = 31.6$ | $2.1596 \times 10^7$ | 15.30% | 98.7756 s |
| | $k_3 = 83.7$ | $1.6765 \times 10^7$ | −12.00% | 117.329 s |

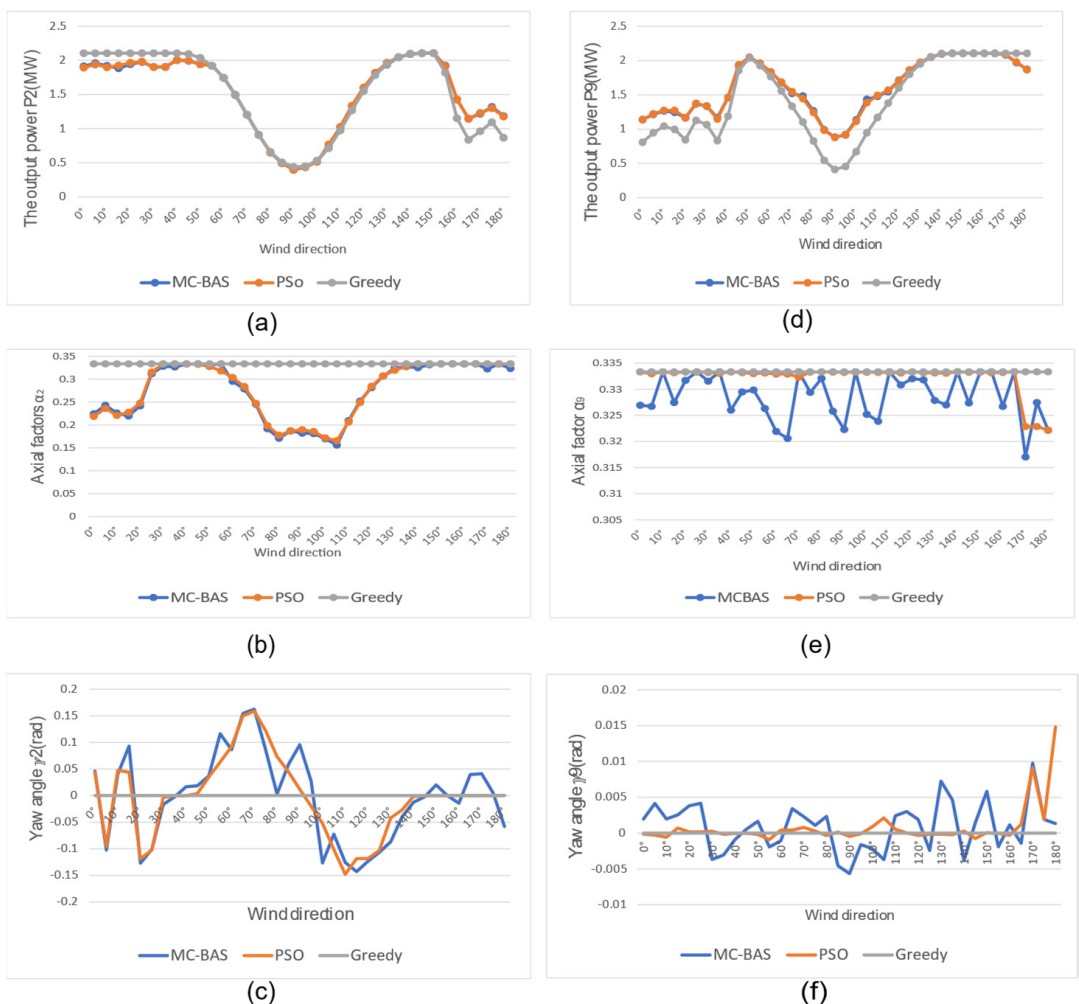

**Figure 7.** Comparison of control parameters of subset N2 between MC-BAS, PSO, and Greedy control with different wind direction: (**a**) The output power of WT2; (**b**) The axial factors of WT2; (**c**) The yaw angle of WT2; (**d**) The output power of WT9; (**e**) The axial factors of WT9; (**f**) The yaw angle of WT9.

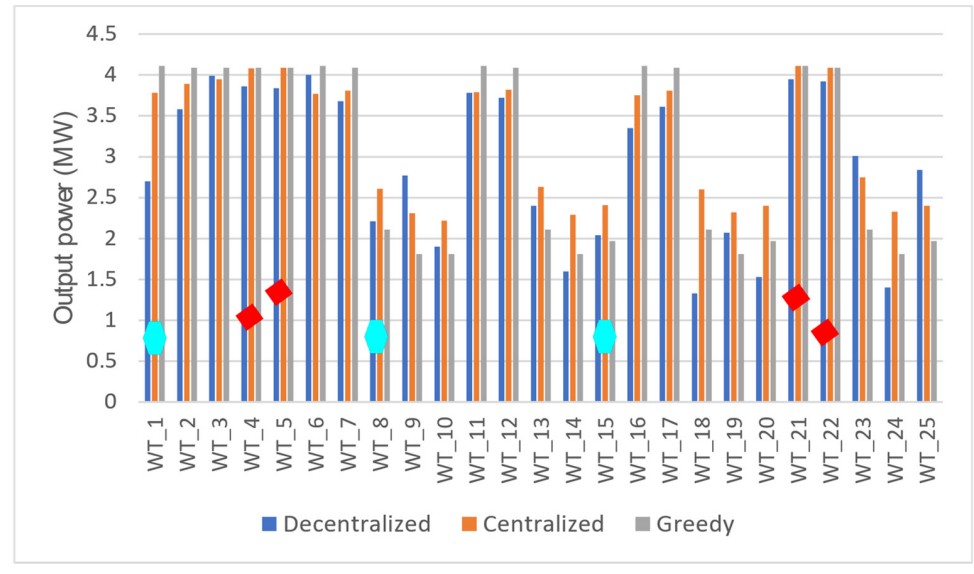

**Figure 8.** The respective power output of wind turbine: $\varphi = 45°$.

**Table 4.** Comparison of power output and calculating time on OWF using centralized and decentralized algorithms on $5 \times 5$ matrix wind farm.

| Wind Direction | Control Method | P_Total (w) | ΔP_Total | T_Total (s) |
|---|---|---|---|---|
| $\varphi = 0°$ | Centralized Greedy | $2.4556 \times 10^7$ | 0.00% | 0.1896 |
| | Centralized MC-BAS | $2.8529 \times 10^7$ | 16.18% | 636.5711 |
| | Decentralized MC-BAS ($k_1 = 0.1$) | $2.8528 \times 10^7$ | 16.17% | 276.245 |
| | Decentralized MC-BAS ($k_2 = 5.6$) | $2.8526 \times 10^7$ | 16.17% | 211.5501 |
| $\varphi = 25°$ | Centralized Greedy | $3.1281 \times 10^7$ | 0.00% | 0.1279 |
| | Centralized MC-BAS | $3.4255 \times 10^7$ | 9.51% | 466.0449 |
| | Decentralized MC-BAS ($k_1 = 1.6$) | $3.4254 \times 10^7$ | 9.50% | 218.4627 |
| | Decentralized MC-BAS ($k_2 = 2.4$) | $3.4252 \times 10^7$ | 9.50% | 138.1018 |
| $\varphi = 45°$ | Centralized Greedy | $3.9268 \times 10^7$ | 0.00% | 0.1102 |
| | Centralized MC-BAS | $4.2164 \times 10^7$ | 7.37% | 286.72145 |
| | Decentralized MC-BAS ($k_1 = 2.5$) | $4.1118 \times 10^7$ | 4.57% | 284.4385 |
| | Decentralized MC-BAS ($k_2 = 5.7$) | $4.0926 \times 10^7$ | 4.16% | 226.5321 |
| $\varphi = 90°$ | Centralized Greedy | $1.8731 \times 10^7$ | 0.00% | 0.1133 |
| | Centralized MC-BAS | $2.3853 \times 10^7$ | 27.35% | 399.0926 |
| | Decentralized MC-BAS ($k_1 = 0.0$) | $2.2795 \times 10^7$ | 21.70% | 112.5742 |
| | Decentralized MC-BAS ($k_2 = 31.6$) | $2.1596 \times 10^7$ | 15.24% | 98.7756 |

Figure 8 presents the results of the respective power output, while Table 4 shows the comparison of total power output. Generally, when the MC-BAS control method is used to implement a centralized and decentralized approach, the total produced power increases compared to the greedy control method. The decoupled cluster subset with wind direction $\varphi = 45°$ can be obtained from Table 4. For example, one cluster subset N1 includes turbines T1, T8, and T15, symbolled as blue hexagon lines. Figure 8 shows that, with the greedy control algorithm, the lead wind turbine T1 produces the maximum power output, while the communication neighbors T8 and T15 only produce minimal power output without regulating the wake effect. Furthermore, the upstream wind turbine can cause significant wake disruption, reducing wind speed and lowering power conversion of the downstream wind turbine [17]. The wake effect is taken into account in the MC-BAS decentralized and centralized control scheme for optimizing overall power output. The majority of upstream turbines reduce output power, whereas downstream turbines increase power conversion, thereby increasing the entire power conversion. Furthermore, subsets N4, N5, N12, N13 have only one turbine, which is symbolized as a little red diamond on WT4, WT5, WT21, and WT22, and, respectively, the output power has no significant difference in the three different control methods since they are unconcerned about the downstream turbine. Other wind directions can be analyzed in the same way. However, the methodologies were not described in this paper to maintain brevity.

The calculated time differs between the decentralized and centralized methods. The control speed of decentralized control is higher than the centralized control because there are fewer turbines to solve, as shown in Table 4. The rate of power (ΔP_total) is the increased power at the baseline of *P*_total of the greedy centralized algorithm. Table 4 shows that ΔP_total increases at varying degrees in the decentralized MC-BAS control and centralized MC-BAS method under different wind directions. Moreover, the decentralized MC-BAS computation time (T_total) is reduced to less than 1/3 times of the centralized approach. The mean total power generated by the centralized MC-BAS algorithm and decentralized MC-BAS algorithm improves by 14.4% and 11.3676%, respectively, compared to the baseline. This indicates about 3.0324% power loss in the decentralized MC-BAS compared to the centralized MC-BAS method. Thus, the proposed control strategy is practical for increasing power output and improving calculation speed from the perspective of real-time control and the profit of the large-scale OWF. For different wind directions,

we set the appropriate value $k_2$ so as to improve the calculation efficiency. Moreover, the importance of the adaptive pruned algorithm is also verified in Table 4.

### 5.3. The Advantage of MC-BAS over Other Algorithms

Generally, the higher the number of iterations, the more accurate the computation. This section aims to take the least number of iterations possible to reach the optimum control actions, resulting in improved total power conversion and communication burden.

According to Figure 9, the calculating time will increase significantly as the number of iterations increases by 100 to 300 in 20 increments, verifying the statements made in Section 3.1. The number of iterations plays a crucial role in reducing the calculation time, therefore reducing the communication burden. Consequently, the exact iteration value is a significant tuning value for optimization algorithms. The MC-BAS algorithm takes far less calculating time than the other three control algorithms and is about 1/9 time of the PSO method and 1/4 time of the GA method. Thus, the proposed centralized MC-BAS method outperforms other intelligent methods (GA and PSO) in terms of calculating speed.

Figure 10 shows the power conversion depending on wind directions and iterations with four different control algorithms. The PSO method can produce more power than others in most iterations. However, when the number of iterations exceeds 140, the total production in the PSO algorithm is equal to that of the MC-BAS algorithm. Consequently, we set the number of iterations to 140 to obtain better total power conversions with the proposed MC-BAS. The main drawback of the GA algorithm is the unstable output power, which varies at different iterations, as shown in Figure 10b,d. Therefore, the GA algorithm is not a suitable choice for OWF. The convergence of the algorithm can be measured by error variation and the number of iterations. In this paper, in order to test the influence of the number of iterations on the results, the number of iterations is used as the condition for the end of the simulation.

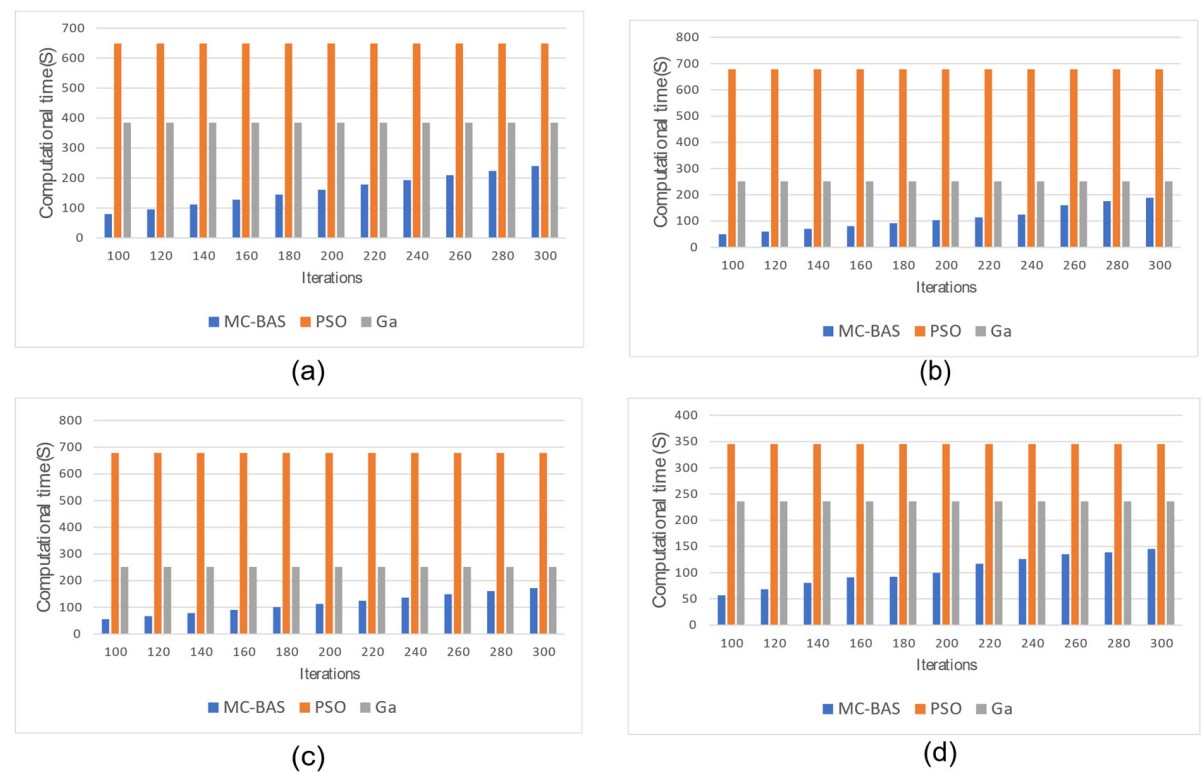

**Figure 9.** The computational time versus iteration of different algorithms in $3 \times 3$ matrix OWF: (**a**) $\varphi = 0°$; (**b**) $\varphi = 25°$; (**c**) $\varphi = 45°$; (**d**) $\varphi = 90°$.

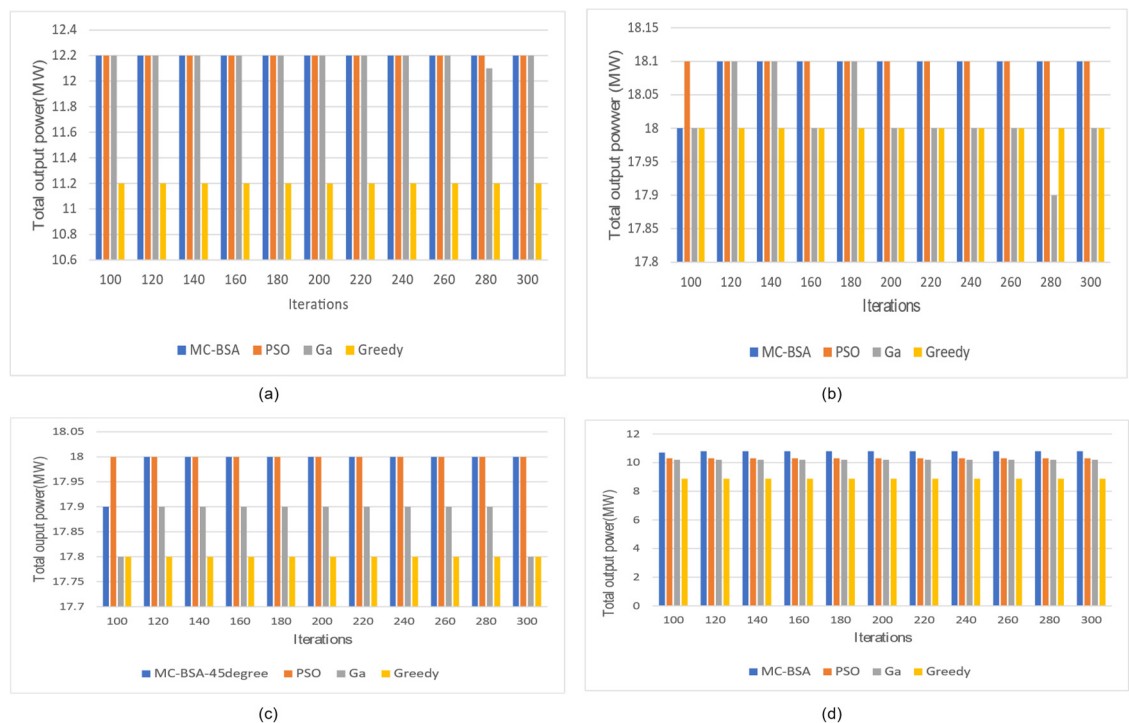

**Figure 10.** The total power output of 3 × 3 matrix OWF on MC-BAS algorithm compared to the iteration algorithms: (**a**) $\varphi = 0°$; (**b**) $\varphi = 25°$; (**c**) $\varphi = 45°$; (**d**) $\varphi = 90°$.

Moreover, we demonstrate the effectiveness and scalability of the proposed methodology. The range of wind direction is $\varphi \in [0°, 10°, \ldots, 180°]$ and the number of turbines is a 2 × 2 matrix and 3 × 3 matrix as shown in Figure 11.

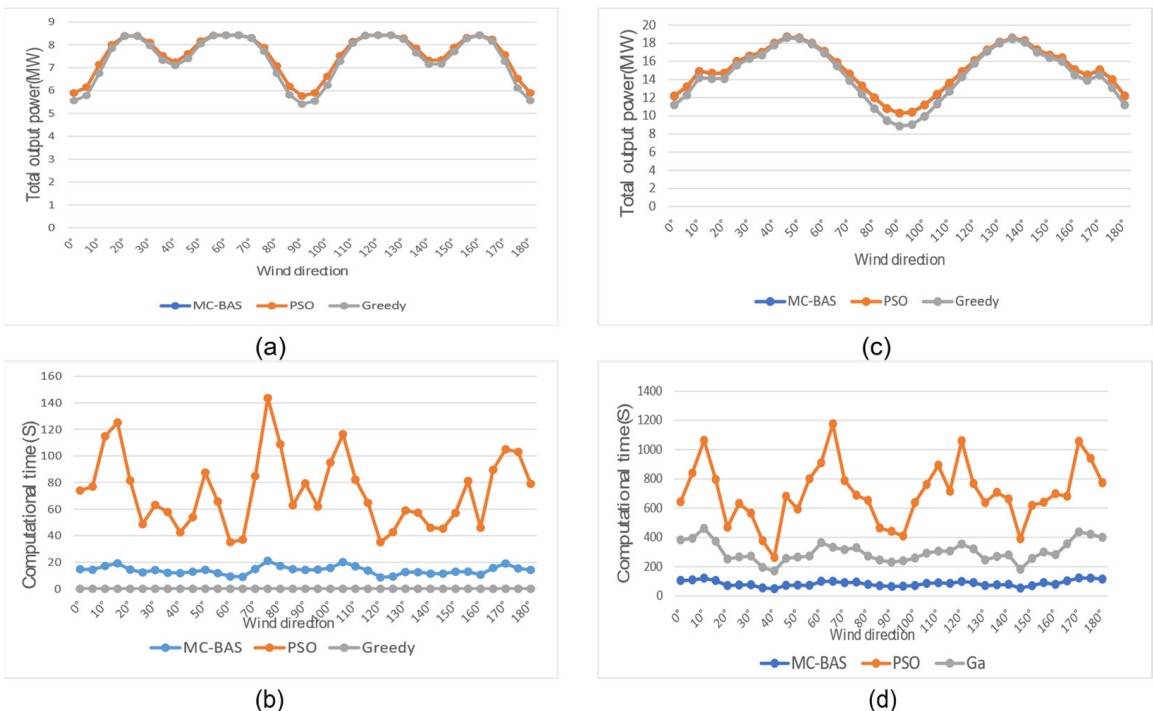

**Figure 11.** Comparison of the centralized MC-BAS algorithms with other iteration methods: (**a**) the total power conversion with 2 × 2 matrix turbines; (**b**) the calculating time with 2 × 2 matrix turbines; (**c**) the total power conversion with 3 × 3 matrix turbines; (**d**) the calculating time with 3 × 3 matrix turbines.

Figure 11a,c shows that the total power conversion of MC-BAS is exactly the same as that of PSO, which is better than the value produced by the greedy algorithm in most wind directions. In addition, the MC-BAS algorithm takes less calculation time than the PSO algorithm on the 2 × 2 matrix and 3 × 3 matrix wind farms (Figure 11b,d). Notably, the ordinate scale of Figure 11c is 10 times that of Figure 11a,d and is 10 times that of Figure 11b. A more detailed analysis with the increasing output is displayed in Tables 5 and 6.

**Table 5.** The total power conversion improvement rate compared to the baseline with the 3 × 3 matrix OWF.

| $\varphi$ | 0° | 5° | 10° | 15° | 20° | 25° | 30° | 35° | 40° | 45° | 50° | 55° | 60° |
|---|---|---|---|---|---|---|---|---|---|---|---|---|---|
| ▲MCBAS | 9% | 7% | 5% | 4% | 4% | 3% | 2% | 2% | 1% | 1% | 1% | 1% | 1% |
| ▲PSO | 9% | 7% | 5% | 4% | 4% | 3% | 2% | 2% | 1% | 1% | 1% | 1% | 1% |
| $\varphi$ | 65° | 70° | 75° | 80° | 85° | 90° | 95° | 100° | 105° | 110° | 115° | 120° | 125° |
| ▲MCBAS | 3% | 5% | 7% | 11% | 14% | 16% | 15% | 13% | 9% | 7% | 3% | 2% | 1% |
| ▲PSO | 3% | 5% | 7% | 11% | 14% | 16% | 15% | 13% | 9% | 7% | 3% | 2% | 1% |
| $\varphi$ | 130° | 135° | 140° | 145° | 150° | 155° | 160° | 165° | 170° | 175° | 180° | | |
| ▲MCBAS | 1% | 0% | 1% | 2% | 2% | 3% | 4% | 4% | 3% | 7% | 9% | | |
| ▲PSO | 1% | 0% | 1% | 2% | 2% | 3% | 4% | 4% | 3% | 7% | 9% | | |

**Table 6.** The total power conversion improvement rate compared to the baseline in 2 × 2 matrix WF.

| $\varphi$ | 0° | 5° | 10° | 15° | 20° | 25° | 30° | 35° | 40° | 45° | 50° | 55° | 60° |
|---|---|---|---|---|---|---|---|---|---|---|---|---|---|
| ▲MCBAS | 6% | 6% | 6% | 2% | 0% | 0% | 2% | 3% | 2% | 3% | 1% | 0% | 0% |
| ▲PSO | 6% | 6% | 6% | 2% | 0% | 0% | 2% | 3% | 2% | 3% | 1% | 0% | 0% |
| $\varphi$ | 65° | 70° | 75° | 80° | 85° | 90° | 95° | 100° | 105° | 110° | 115° | 120° | 125° |
| ▲MCBAS | 0% | 0% | 2% | 5% | 7% | 7% | 6% | 6% | 3% | 1% | 0% | 0% | 0% |
| ▲PSO | 0% | 0% | 2% | 5% | 6% | 6% | 6% | 6% | 3% | 1% | 0% | 0% | 0% |
| $\varphi$ | 130° | 135° | 140° | 145° | 150° | 155° | 160° | 165° | 170° | 175° | 180° | | |
| ▲MCBAS | 1% | 3% | 2% | 2% | 2% | 1% | 0% | 1% | 4% | 7% | 6% | | |
| ▲PSO | 1% | 3% | 2% | 2% | 2% | 1% | 0% | 1% | 4% | 7% | 6% | | |

Tables 5 and 6 show the total production improvement rate is relative to the baseline of greedy control with 2 × 2 matrix turbines and 3 × 3 matrix turbines. It can be observed in some wind directions that the increased power rate is zero, such as $\varphi = 20°, 25°$, etc., in Table 6 and $\varphi = 135°$ in Table 5. The advantage of the proposed MC-BAS algorithm was verified by comparing it with several other algorithms, especially in increasing the output power and decreasing the calculating time. Additionally, in a large-scale OWF, an adaptive pruned wake digraph is proposed to divide it into several decoupled subsets. Then, the same controller works on every subset to ensure real-time control. By analyzing the data in Tables 3–6, we can conclude that the ΔP with the MC-BAS algorithm increases with a greater number of wind turbines on the wind farm. For example, when $\varphi = 15°$, ΔP = 2%, 4%, 12.73% on the 2 × 2, 3 × 3, and 5 × 5 matrix wind farms, respectively. We anticipate that the proposed algorithm will demonstrate a good performance of large-scale wind farms.

## 6. Conclusions

This paper proposed a decentralized real-time power optimization for large-scale OWFs using an adaptive pruned wake digraph approach. The results of this paper can be summarized as follows:

1.  The proposed adaptive pruning algorithm fully considers the real-time power optimization control goals, providing a suitable method of grouping to avoid obtaining a sub-optimization result due to the unsuitable communication topology. The vital point of the adaptive pruned digraph is to uncover the accurate global threshold $\varepsilon_p$ corresponding to the different wind by setting the suitable parameter $k_2$. Moreover, the proposed method was verified to be efficient by the Simulink result, and the off-line look-up table was constructed in Appendix B.

2.  This work presents a modified BAS algorithm to raise BAS's ability and efficiency for dealing with high-dimensional nonlinear problems. The BAS can use fewer iterations to rapidly search for the fitness function maximum in the parameter selection space. Meanwhile, the Monte Carlo (MC) law of Simulate Anneal (SA) was introduced to improve the reproducibility and stability of the algorithm by avoiding blind searching and escaping the local traps minima.

3.  For a large-scale wind farm, real-time state information may be excessive for the high communication and computational burden—centralized control approaches might fail. However, the adaptive pruned digraph decentralized operation can solve this problem by dividing the large-scale wind farm into several decoupled subsets; the local controller only deals with the local subset.

Future work will focus on increasing the control parameters and control objectives of the large-scale OWF, considering the infection of nonlinear turbulent flow [50,51]. Moreover, optimizing the wind farm layout with irregularly shaped wind farms will be studied by decreasing the wake effect.

**Author Contributions:** Conceptualization, Y.-H.J.; methodology, Y.C.; software, Y.C.; validation, Y.-H.J.; investigation, D.S.; data curation, Y.C.; writing—original draft preparation, Y.C.; writing—review and editing, D.S.; visualization, D.S.; supervision, Y.-H.J. All authors have read and agreed to the published version of the manuscript.

**Funding:** This work was partially supported by the Basic Science Research Program through the National Research Foundation of Korea (NRF), funded by the Ministry of Education (NRF-2016R1A6A1A03013567, NRF-2021R1A2B5B01001484) and the Innovation-Driven Project of Central South University (2020CX031).

**Institutional Review Board Statement:** Not applicable.

**Conflicts of Interest:** The authors declare no conflict of interest.

## Appendix A

**Table A1.** The main parameters of the 5 MW wind turbine for an offshore wind farm.

| P_rate | 5 MW |
|---|---|
| D | 126 m |
| $\omega^{min}$ | 6.9 rpm |
| $\omega^{max}$ | 12.1 rpm |
| $\beta^{max}$ | 90° |
| Gearbox ratio | 97:1 |
| Rated wind speed | 11.4 m/s |
| $C_p^{max}$ | 0.485 |
| Hub height | 90 m |

## Appendix B

**Table A2.** The look-up table of different $k_2$ with the varying wind.

| $V_\infty$ | $\varphi$ | 0° | 15° | 30° | 45° | 60° | 75° | 90° | 105° | 120° | 135° | 150° | 165° | 180° |
|---|---|---|---|---|---|---|---|---|---|---|---|---|---|---|
| 8 m/s | $k_2$ | 5.6 | 2.4 | 3.5 | 4.8 | 6.8 | 16.5 | 31.6 | 19.9 | 7.5 | 5.3 | 4.3 | 3.8 | 6.2 |
| 9 m/s | $k_2$ | 5.9 | 2.8 | 3.9 | 5.1 | 7.4 | 16.9 | 31.8 | 21.3 | 8.0 | 5.7 | 4.5 | 5.3 | 6.7 |
| 10 m/s | $k_2$ | 6.2 | 3.3 | 4.2 | 5.7 | 7.6 | 17.6 | 41.8 | 27.3 | 9.6 | 6.7 | 5.8 | 6.9 | 7.6 |
| 11 m/s | $k_2$ | 6.8 | 3.9 | 6.4 | 8.4 | 18.9 | 43.4 | 29.8 | 10.9 | 7.9 | 6.9 | 7.3 | 7.6 | 9.4 |

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
