# Peer review of "Modified Beetle Annealing Search (BAS) Optimization Strategy for Maxing Wind Farm Power through an Adaptive Wake Digraph Clustering Approach"

_energies, doi:10.3390/en14217326_

Round 1
Reviewer 1 Report
The paper aims to present a new method to optimize yaw angles and axial induction factors in a large offshore wind farm to maximize power generation. The authors propose a decentralized non-convex optimization strategy based on a modified beetle antennae search (BAS) algorithm. The algorithm works in such a way that it can split the wind farm into nearly uncoupled clustering communication subsets. A Monte Carlo-based beetle annealing optimization strategy is then used to optimize yaw angles and axial induction factors in every subgroup of turbines.
This investigation is certainly relevant and could be of interest to a wide part of the wind energy community. The methodology is robust, and the conclusions are supported by the results. The authors are however asked to address the following comments in the revised submission:
1 – In the first place, I would encourage the authors to extend the abstract by including a more compelling motivation and key results. The motivation does not fully explain what the problem is and why it should be solved more efficiently (“The centralized control strategy may be inefficient due to the significant computing burden and communication dependency in a large offshore wind farm”). Also, the abstract does not quite convey the interesting results that follow in the main paper (“Finally, the simulation results demonstrated that the proposed approach is practical in reducing the computational cost, allowing it for nonlinear and real-time operations in the OWF”).
2 – The readability and presentation of the study should be further improved. The paper suffers from language problems. The paper should be proofread by a native speaker or a proofreading agent.
3 – There have been a lot of advancements on wake models for wind farm control and layout optimization. Wake models based on computational fluid dynamics are becoming (and will be in the near future) the state-of-the-art. See for example Ref. [1-4]. The author should mention these, also for potential implementation in their framework in the future.
4 – I have recently read a paper [5] that in a similar manner uses a graph to capture the spatial dependencies caused by wake interactions. That paper was focused on wind farm layout optimization as opposed to control optimization (yaw angle and axial induction factor). It seems that graphs can be efficiently used for both applications, and we should stress more these potential advantages. Can the authors comment on this by highlighting similarities and differences of the two applications?
5 – The formulated optimization problem should be explained in a physic level instead of mathematical way. Specifically, more discussions should be added to Sec. 3 and 4, so the readers can understand its underlying meaning related to the control problem.
6 – Why did the authors select only one wind speed and wind directions ranging from 0 to 180? Please give a motivation.
7 – In all the figures where there is a graph, the values along the edges are difficult to read. A possible solution would be to remove those values and use the line width as a proxy for the edge value. For example, if the max line width is equal to 10, you would make the actual line width proportional to the edge value in the range 0-10. Thicker lines would then represent stronger wake interactions.
8 – What are the symbols in Fig 8? I see blue hexagon-line and yellow diamond-like markers that are not explained.
9 – In all the figure there you plot output power, please use MW (megawatts) instead of W (watts) as the unit of measure. In this way, you avoid writing E+06 or six 0 in the axis values.
10 – In all the figure there you plot the computational time, you have a typo. “Calculating t me” should be “Calculating time”. However, I would suggest using “Computational time”.
References
[1] Kuo et al., "Wind farm layout optimization on complex terrains – Integrating a CFD wake model with mixed-integer programming," Applied Energy, 2016.
[2] King et al., "Adjoint optimization of wind farm layouts for systems engineering analysis," Proceedings of the 34th Wind Energy Symposium, 2016.
[3] King et al., "Optimization of wind plant layouts using an adjoint approach," Wind Energy Science, 2017.
[4] Serrano González et al., “A review and recent developments in the optimal wind-turbine micro-siting problem”, Renewable and Sustainable Energy Reviews, 2014.
[5] Dhoot et al., “Optimizing wind farms layouts for maximum energy production using probabilistic inference: Benchmarking reveals superior computational efficiency and scalability”, Energy, 2021
Reviewer 2 Report
Summary
The paper presents a novel approach to controlling an array of wind turbines. In the presented approach all turbines are segregated into groups using a pruned graph that identifies turbines which are aerodynamically strongly coupled. In a second step the optimal yaw angle of each group is determined by a beetle antennae search optimization algorithm in order to maximize the overall power output of the entire array. The optimization algorithm tolerates non-optimal results with a certain probability in order to prevent convergence to local optima. The method delivers results that are as good as or better than classical approaches at an overall reduction of computational effort.
Mandatory changes
Overall, the paper is written in a concise, complete way and is easy to understand. However, some of the concepts might be explained in more detail in order to allow for a better understanding of the paper. The authors are therefore invited to incorporate the following points into the current version of the paper:
- 52: consider explaining the terms "centralized control", "distributed control" and "cooperate control" in more detail for uninformed readers new to the topic (What is controled? What is the distinction b/w the approaches).
- 60: What is the one dimension? A spatial coordinate (distance of the turbines)? What is additional variable in two-dimensional data? Spacing in the same row of turbines? Please elaborate to help understand this issue w/o having to read the references.
- 140: Is (x,y,z) really the "direction" of V? In other words: Is V really a vector? I assume V to rather be a scalar and (x,y,z) to be the spatial location/position (not direction) of V. Please check!
- 140: Please replace "delta is the wake centerline" with a more elaborate description such as "lateral deflection of wake centerline" and reference the Fig 1.
- 154/155: Show delta_0 and x_0 in Fig. 1.
- 157: Clarify the difference of "the turbine model" and the previously presented wake model.
- Eqn 6: Add indices i,j to A_overlap
- 197: Please clarify if and how interaction of different upstream turbines on a downstream turbine is resolved in the model. Does one turbine see the sum of the wakes of all upstream turbines or are these analyzed in isolation?
- 222: Define phi
- 277: Give a short, intuitive explanation of the overall steps and nomenclature of the BAS algorithm for the uninformed reader (such as "1. beetle moves in best direction, 2. antennae search for new best direction, 3. repeat") and inherent pros and cons wrt other algorithms.
- Eqn 9: Why is the contraint on power necessary? May the model produce unphysical values (such as negative powers) if alpha any gamma are in the specified ranges?
- Eqn 9: As I understand, Algorithm I is looking for a minimum (lines 295, 296) whereas eqn 9 declares max(f) to be the optimum. Please comment!
- 376: Is it really lowered? There should be a smaller overlap area of wakes/downstream turbines and thus a higher power output.
- 439: This seems to be true only for large incidence angles. Please be more specific.
- Fig 10: The GA results fluctuate a lot for different numbers of iterations in (b) and (c) and probably ran into different local optima. Please comment on this! How was convergence of the algorithm checked?
- 672: Please provide a proper reference for 41
Optional changes
Furthermore, some adjustments to language and style need to be made. Attached is a list of recommendations.
- consider replacing "power/energy production/generation" -> "power/energy conversion" (and similar) as per thermodynamic principles energy and power are not produced but merely converted
- 46: consider removing "been"
- 51: consider removing "very"
- 74: consider removing "As we all know"
- 79: consider replacing "on analytical" -> "on an analytical"
- 81: consider replacing "is" -> "are"
- 89: consider replacing "get global" -> "get a global"
- 129: what is meant by "vale angle"?
- 138: remove "the following" (eqn 1 is not the NSE)
- 141, 357: replace "desperately" -> "respectively"
- 157: consider removing "As we know"
- 217: consider replacing "decoupled topology" -> "a decoupled topology"
- 223: consider removing "As we know"
- algorithm I, step 4: replace "defined" -> "define"
- algorithm I, step 4: replace "each" -> "for each"
- algorithm I, step 5: replace "tun" -> "to tune"
- algorithm I, step 7: replace "tun" -> "tune"
- algorithm I, step 8: replace "analysis" -> "analyze"
- 233: replace "between" -> "in"
- 239: consider replacing "look" -> "look-up"
- Fig. 3: typo in "transmation"?
- Algorithm 1: Should be distinguished form "algorithm I"
- Algorithm 1: All references to equations need to be updated!
- Algorithm 1: Why are there different indentations b/w lines 292/293, 294/295, 299/300, etc.?
- Algorithm 1: Consider using a different symbol for "P" to avoid confusion with power (also in eqn 14, etc.)
- Algorithm 1: in line 293, replace "x_1" (x_one) -> "x_l" (x_L)
- 157: consider replacing "diffused" -> "diffusion"
- Tab 1: replace "NO" -> "No"
- 399: replace "that" -> "of"
- Tab 3: replace "P(w)" -> "P(W)"
- Fig 7 - 11: Consider removing "The" from y axis titles and use the same legend entries where appropriate
- Fig 8: The symbols seem to be misplaced
- Tab 2, 3, 4: Distinguish T (time) from T in Algorithm 1
- 471: replace "symbolled" -> "symbolized"
- 527: replace "comparable to" -> "exactly the same as"
- 536: remove "is"
- 565: replace "improves" -> "improve"
Round 2
Reviewer 1 Report
The authors have suitably addressed my comments. The paper can be accepted for publication.